# Differences between Very Highly Sensitized Kidney Transplant Recipients as Identified by Machine Learning Consensus Clustering

**DOI:** 10.3390/medicina59050977

**Published:** 2023-05-18

**Authors:** Charat Thongprayoon, Jing Miao, Caroline C. Jadlowiec, Shennen A. Mao, Michael A. Mao, Pradeep Vaitla, Napat Leeaphorn, Wisit Kaewput, Pattharawin Pattharanitima, Supawit Tangpanithandee, Pajaree Krisanapan, Pitchaphon Nissaisorakarn, Matthew Cooper, Wisit Cheungpasitporn

**Affiliations:** 1Division of Nephrology and Hypertension, Department of Medicine, Mayo Clinic, Rochester, MN 55905, USA; thongprayoon.charat@mayo.edu (C.T.); miao.jing@mayo.edu (J.M.); supawit_d@hotmail.com (S.T.); pajaree_fai@hotmail.com (P.K.); 2Division of Transplant Surgery, Mayo Clinic, Phoenix, AZ 85054, USA; 3Division of Transplant Surgery, Mayo Clinic, Jacksonville, FL 32224, USA; 4Division of Nephrology and Hypertension, Department of Medicine, Mayo Clinic, Jacksonville, FL 32224, USA; 5Division of Nephrology, University of Mississippi Medical Center, Jackson, MS 39216, USA; 6Department of Military and Community Medicine, Phramongkutklao College of Medicine, Bangkok 10400, Thailand; 7Department of Internal Medicine, Faculty of Medicine, Thammasat University, Pathum Thani 12120, Thailand; 8Department of Medicine, Division of Nephrology, Massachusetts General Hospital, Harvard Medical School, Boston, MA 02114, USA; 9Medical College of Wisconsin, Milwaukee, WI 53226, USA

**Keywords:** clustering, highly sensitized kidney transplant recipients, kidney transplant, kidney transplantation, transplantation

## Abstract

*Background and Objectives*: The aim of our study was to categorize very highly sensitized kidney transplant recipients with pre-transplant panel reactive antibody (PRA) ≥ 98% using an unsupervised machine learning approach as clinical outcomes for this population are inferior, despite receiving increased allocation priority. Identifying subgroups with higher risks for inferior outcomes is essential to guide individualized management strategies for these vulnerable recipients. *Materials and Methods*: To achieve this, we analyzed the Organ Procurement and Transplantation Network (OPTN)/United Network for Organ Sharing (UNOS) database from 2010 to 2019 and performed consensus cluster analysis based on the recipient-, donor-, and transplant-related characteristics in 7458 kidney transplant patients with pre-transplant PRA ≥ 98%. The key characteristics of each cluster were identified by calculating the standardized mean difference. The post-transplant outcomes were compared between the assigned clusters. *Results*: We identified two distinct clusters and compared the post-transplant outcomes among the assigned clusters of very highly sensitized kidney transplant patients. Cluster 1 patients were younger (median age 45 years), male predominant, and more likely to have previously undergone a kidney transplant, but had less diabetic kidney disease. Cluster 2 recipients were older (median 54 years), female predominant, and more likely to be undergoing a first-time transplant. While patient survival was comparable between the two clusters, cluster 1 had lower death-censored graft survival and higher acute rejection compared to cluster 2. *Conclusions*: The unsupervised machine learning approach categorized very highly sensitized kidney transplant patients into two clinically distinct clusters with differing post-transplant outcomes. A better understanding of these clinically distinct subgroups may assist the transplant community in developing individualized care strategies and improving the outcomes for very highly sensitized kidney transplant patients.

## 1. Introduction

Kidney transplantation is recognized as the most effective treatment for enhancing the quality of life and survival in patients with end-stage kidney disease (ESKD) [1,2]. However, the waitlist for a kidney transplant can be lengthy or unproductive for highly sensitized patients who are unable to find an immunologically compatible donor [3]. Compared to unsensitized patients, highly sensitized kidney transplant recipients often have poorer allograft clinical outcomes and patient survival [4]. The production of alloantibodies against human leukocyte antigens (HLAs), generated through blood transfusions, previous transplants, infections, and pregnancy, causes sensitization. The panel reactive antibody (PRA) test is commonly used to determine the sensitization status of potential kidney transplant candidates, and those with a PRA of ≥80% are typically considered highly sensitized [5], and candidates with a PRA ≥ 98% are considered very highly sensitized and receive increased allocation priority [6]. Although approximately 30% of waitlist kidney transplant candidates are sensitized, only 6.5% receive a transplant each year [7]. In Europe, around 20% of patients waiting for a kidney transplant are sensitized, with 5% being highly sensitized [8].

Machine learning (ML) is an artificial intelligence subfield that can effectively analyze large datasets autonomously, and has been successfully applied in clinical medicine [9,10,11]. ML algorithms can be categorized into three main types: supervised learning (such as classification and regression), unsupervised learning (such as clustering, association, and dimensionality reduction), and reinforcement learning [12,13]. Consensus clustering is a technique in ML that merges the outcomes of various clustering algorithms to identify stable and robust clusters within a dataset [12,13]. It addresses the problem of instability and inconsistency in traditional clustering methods by forming a consensus partition that offers a more dependable and consistent clustering solution, capturing the fundamental data structure and lowering the risk of identifying false or unreliable clusters [12,13].

Kidney transplant recipients are a heterogeneous population with distinct underlying illnesses, genetic profiles, and immune responses. This variability can lead to inconsistent outcomes after transplantation, with some patients achieving exceptional graft functionality while others encounter issues such as rejection or allograft failure. Recent research has demonstrated unique subgroups identified by ML consensus clustering among solid organ transplant recipients such as the heart, kidney, and liver [14,15,16,17,18,19]. After identifying patient subgroups, researchers can assess the contrast in clinical results including graft survival, rejection rates, and infections among the various subgroups. This knowledge can guide clinical decision-making and help develop customized treatment plans for each patient based on their subgroup classification.

This cohort study aims to categorize very highly sensitized kidney transplant recipients with a PRA ≥ 98% based on comprehensive recipient-, donor-, and transplant-related variables using an unsupervised ML tool, and then assess post-transplant outcomes among the ML identified distinct clusters.

## 2. Materials and Methods

### 2.1. Data Source and Study Population

Data for this study were sourced from the Organ Procurement and Transplantation Network (OPTN)/United Network for Organ Sharing (UNOS) database for patients who underwent kidney transplants in the United States between 2010 and 2019. Very highly sensitized kidney transplant patients, defined as having pre-transplant PRA ≥ 98% [6], were included, while patients who received simultaneous kidney transplants with other organs were excluded. The Mayo Clinic Institutional Review Board (IRB number 21-007698) approved this study.

### 2.2. Data Collection

The OPTN/UNOS database contains patient-level data of all transplant events in the United States. The recipient-, donor-, and transplant-related variables, listed in Table 1, were abstracted from the OPTN/UNOS database to be included in the ML cluster analysis. All variables had less than 10% missing data (Appendix A), and missing data were imputed using the multivariable imputation by the chained equation (MICE) method [20].

### 2.3. Clustering Analysis

In this study, the clinical phenotypes of very highly sensitized kidney transplant patients were categorized using an unsupervised ML technique called consensus clustering [21]. We utilized a standard subsampling parameter of 80%, with 100 iterations and a range of potential clusters from 2 to 10, to perform the consensus clustering analysis. This approach aimed to prevent generating an excessive number of clusters that would not have clinical significance. The optimal number of clusters was determined by assessing several metrics including the consensus matrix (CM) heat map, cumulative distribution function (CDF), cluster-consensus plots with within-cluster consensus scores, and the proportion of ambiguously clustered pairs (PAC). We used the within-cluster consensus score to evaluate the stability of each cluster. This score ranges from 0 to 1 and is defined as the average consensus value for all pairs of individuals belonging to the same cluster, with a score closer to 1 indicating better cluster stability. Additionally, we calculated the PAC as the proportion of all sample pairs, with consensus values falling within the predetermined boundaries. A value closer to 0 indicates better cluster stability [12]. For reproducibility, the detailed consensus cluster algorithms used in this study are available in the Appendix A. The workflow of consensus clustering is shown in Figure 1.

### 2.4. Outcomes

The research investigated various outcomes in kidney transplant recipients such as patient death, death-censored graft failure at 1 and 5 years, and acute allograft rejection within one year of transplantation. Nevertheless, it is crucial to mention that the OPTN/UNOS registry merely offers information about whether allograft rejection happened within a year following the kidney transplant, without specifying the exact date of occurrence.

### 2.5. Statistical Analysis

We utilized the consensus clustering approach to categorize highly sensitized kidney transplant patients and compared their clinical characteristics and post-transplant outcomes among the assigned clusters. Categorical and continuous characteristics were compared using chi-squared tests and analysis of variance, respectively. To identify key characteristics of each cluster, we looked for a standardized mean difference of >0.3 compared to the overall cohort. We then used the Kaplan–Meier method to estimate cumulative risks of patient survival, death-censored graft survival, and overall graft survival following kidney transplant. The log-rank test was employed to compare the assigned clusters, while Cox proportional hazard regression was used to estimate hazard ratios for patient death, death-censored graft failure, and overall graft failure. Conversely, to compare the incidence of 1-year acute allograft rejection, we used the chi-squared test, and logistic regression was used to estimate the odds ratio. Statistical analyses were performed using R, version 4.0.3, with the Consensus-ClusterPlus package (version 1.46.0) utilized for consensus clustering analysis. This allowed us to determine the optimal number of clusters by examining the CM heat map, CDF, cluster-consensus plots with the within-cluster consensus scores, and the PAC. Finally, we used the MICE command in R for multivariable imputation by chained equation [20].

## 3. Results

Out of the 158,367 adult patients receiving kidney transplants from 2010 to 2019 in the United States, 7458 (4.7%) were very highly sensitized. Thus, we performed consensus clustering analysis in a total of 7458 very highly sensitized kidney transplant patients. Of these, 1081 (14%), 2129 (29%), and 4248 (57%) had a PRA of 98, 99, and 100%, respectively. Table 1 shows the recipient-, donor-, and transplant-related characteristics of the included patients.

**Table 1 medicina-59-00977-t001:** Clinical characteristics according to clusters of very highly sensitized kidney transplant recipients.

	All (*n* = 7458)	Cluster 1 (*n* = 4105)	Cluster 2 (*n* = 3353)	*p*-Value
Recipient age (year), median (IQR)	49 (39–58)	45 (35–54)	54 (45–62)	<0.001
Recipient male sex	2697 (36)	2409 (59)	288 (9)	<0.001
Recipient race - White - Black - Hispanic - Other	2979 (40) 2321 (31) 1486 (20) 672 (9)	1803 (44) 1249 (30) 736 (18) 317 (8)	1176 (35) 1072 (32) 750 (22) 355 (11)	<0.001
Body mass index (kg/m^2^), median (IQR)	26.7 (23.0–31.2)	25.7 (22.5–30.0)	28.0 (24.0–32.4)	<0.001
Retransplant	4139 (56)	4032 (98)	107 (3)	<0.001
Dialysis duration - Preemptive - <1 year - 1–3 years - >3 years	713 (10) 644 (9) 2037 (27) 4064 (54)	240 (6) 299 (7) 1098 (27) 2468 (60)	473 (14) 345 (10) 939 (28) 1596 (48)	<0.001
Cause of end-stage kidney disease - Diabetes mellitus - Hypertension - Glomerular disease - PKD - Other	1095 (15) 1072 (14) 1452 (19) 456 (6) 3383 (45)	130 (3) 338 (8) 690 (17) 105 (3) 2842 (69)	965 (29) 734 (22) 762 (23) 351 (10) 541 (16)	<0.001
Comorbidity - Diabetes mellitus - Malignancy - Peripheral vascular disease	1858 (25) 602 (8) 532 (7)	686 (17) 342 (8) 232 (6)	1172 (35) 260 (8) 300 (9)	<0.001 0.363 <0.001
PRA (%) - 98 - 99 - 100	1081 (15) 2129 (29) 4248 (57)	449 (11) 1010 (25) 2646 (65)	632 (19) 1119 (33) 1602 (48)	<0.001
Positive HCV serostatus	344 (5)	193 (5)	151 (5)	0.685
Positive HBs antigen	101 (1)	62 (2)	39 (1)	0.197
Positive HIV serostatus	28 (0)	20 (0)	8 (0)	0.081
Functional status - 10–30% - 40–70% - 80–100%	8 (0) 3374 (45) 4076 (55)	4 (0) 1761 (43) 2340 (57)	4 (0) 1613 (48) 1736 (52)	<0.001
Working income	2227 (30)	1361 (33)	866 (26)	<0.001
Public insurance	5730 (77)	3313 (81)	2417 (72)	<0.001
U.S. resident	7368 (99)	4078 (99)	3290 (98)	<0.001
Undergraduate education or above	4035 (54)	2338 (57)	1697 (51)	<0.001
Serum albumin (g/dL), mean (SD)	3.9 ± 0.6	3.9 ± 0.6	3.9 ± 0.6	0.483
Kidney donor status - Non-ECD deceased - ECD deceased - Living	6508 (87) 384 (5) 566 (8)	3633 (89) 185 (5) 287 (7)	2875 (86) 199 (6) 279 (8)	0.001
ABO incompatibility	13 (0)	9 (0)	4 (0)	0.303
Donor age (year), median (IQR)	35 (24–47)	33 (23–45)	36 (25–47)	<0.001
Donor male sex	4528 (61)	2549 (62)	1979 (59)	0.007
Donor race - White - Black - Hispanic - Other	4676 (63) 1158 (16) 1279 (17) 345 (5)	2605 (63) 617 (15) 694 (17) 189 (5)	2071 (62) 541 (16) 585 (17) 156 (5)	0.454
History of hypertension in donor	1409 (19)	712 (17)	697 (21)	<0.001
KDPI - Living donor - KDPI < 85 - KDPI ≥ 85	566 (8) 6759 (91) 133 (2)	287 (7) 3764 (92) 54 (1)	279 (8) 2995 (89) 79 (2)	<0.001
HLA mismatch, median (IQR)	3 (2–4)	3 (1–4)	3 (2–4)	<0.001
Cold ischemia time (hour), median (IQR)	18.5 (13.2–23.8)	19.0 (13.9–24.1)	17.8 (12.3–23.2)	<0.001
Kidney on pump	2123 (28)	1146 (28)	977 (29)	0.245
Delay graft function	1900 (25)	1248 (30)	652 (19)	<0.001
Allocation type - Local - Regional - National	2211 (30) 1729 (23) 3518 (47)	1070 (26) 886 (22) 2149 (52)	1141 (34) 843 (25) 1369 (41)	<0.001
EBV status - Low risk - Moderate risk - High risk	48 (1) 6734 (90) 676 (9)	26 (1) 3713 (90) 366 (9)	22 (1) 3021 (90) 310 (9)	0.878
CMV status - D−/R− - D−/R+ - D+/R+ - D+/R−	841 (11) 2135 (29) 1085 (15) 3397 (46)	508 (12) 1192 (29) 663 (16) 1742 (42)	333 (10) 943 (28) 422 (13) 1655 (49)	<0.001
Induction immunosuppression - Thymoglobulin - Alemtuzumab - Basiliximab - Other - No induction	2432 (73) 1125 (15) 468 (6) 91 (1) 490 (7)	3035 (74) 633 (15) 221 (5) 60 (1) 258 (6)	2432 (73) 492 (15) 247 (7) 31 (1) 232 (7)	0.173 0.370 <0.001 0.036 0.272
Maintenance Immunosuppression - Tacrolimus - Cyclosporine - Mycophenolate - Azathioprine - mTOR inhibitors - Steroid	6937 (93) 84 (1) 6972 (93) 17 (0) 35 (0) 5995 (80)	3789 (92) 62 (2) 3806 (93) 8 (0) 21 (1) 3345 (81)	3148 (94) 22 (1) 3166 (94) 9 (0) 14 (0) 2650 (79)	0.008 0.001 0.003 0.508 0.554 0.008

Abbreviations: BMI: Body mass index, CMV: Cytomegalovirus, D: Donor, EBV: Epstein-Barr virus, ECD: Extended criteria donor, HBs: Hepatitis B surface, HCV: Hepatitis C virus, HIV: Human immunodeficiency virus, KDPI: Kidney donor profile index, mTOR: Mammalian target of rapamycin, PKD: Polycystic kidney disease, PRA: Panel reactive antibody, R: Recipient.

In Figure 2A, the CDF plot consensus distribution for each cluster of highly sensitized kidney transplant patients is displayed. The plot shows that the largest changes in the area occurred between k = 2 and k = 5 (Figure 2B), after which the relative increase in the area became smaller. Figure 2C and Appendix A show the CM heat map, which indicates that the ML algorithm identified cluster 2 with clear boundaries, suggesting that the cluster is stable across repeated iterations. Figure 3A shows that the mean cluster consensus score was the highest in cluster 2, indicating greater cluster stability. Additionally, Figure 3B displays favorable low PAC for two clusters. Therefore, the consensus clustering analysis utilized baseline variables at the time of transplant to identify two clusters that best represented the data pattern of very highly sensitized kidney transplant recipients in this study.

### 3.1. Clinical Characteristics Based on Clusters of Very Highly Sensitized Kidney Transplant Patients

Two distinct clinical clusters were identified by the consensus clustering analysis, as illustrated in Table 1. Cluster 1 was composed of 4105 (55%) patients, while cluster 2 had 3353 (45%) patients. Figure 4 presents the primary features for each cluster, represented by the plot of standardized mean differences.

Cluster 1 patients were younger (median age 45 (IQR 35–54)), more likely male (59% vs. 9%), and more likely to be undergoing kidney re-transplantation (98% vs. 3%). In comparison, cluster 2 recipients were older (median age 54 (45–62) years), more likely to be female (91% vs. 41%), and more likely to be undergoing a first-time transplant (97% vs. 2%). Additionally, more patients in cluster 1 had a PRA of 100% (65% vs. 48%) and had greater than 3 years of dialysis time prior to undergoing transplant (60% vs. 48%). Recipients in cluster 1 also experienced longer cold ischemia time (median (IQR), 19 (13.9–24.1) vs. 17.8 (12.3–23.2) hours), and were more likely to have received a nationally allocated kidney (52% vs. 41%) in comparison to cluster 2.

### 3.2. Post-Transplant Outcomes Based on Clusters of Very Highly Sensitized Kidney Transplant Patients

Table 2 shows the cluster-based post-transplant outcomes. The 1-year and 5-year patient survival was 97.3% and 88.0% in cluster 1, and 97.1% and 86.5% in cluster 2, respectively (*p* = 0.19) (Figure 5A). The 1-year and 5-year death-censored graft survival was 96.3% and 84.4%% in cluster 1, and 97.6% and 85.5% in cluster 2, respectively (*p* = 0.006) (Figure 5B). Cluster 1 had lower death-censored graft survival compared to cluster 2 (HR 1.57, 95% CI 1.18–2.09 and HR 1.28, 95% CI 1.07–1.57 at 1 and 5 years, respectively). Cluster 1 had a higher incidence of acute rejection compared to cluster 2 (9.2% vs. 5.4%; *p* < 0.001).

## 4. Discussion

Inferior clinical outcomes have been observed in very highly sensitized kidney transplant recipients (PRA ≥ 98%). Factors contributing to these inferior outcomes are multifactorial and include variables both directly and indirectly linked to immunologic risk [1,8,22,23]. Using an unsupervised ML consensus clustering tool, we were able to categorize very highly sensitized kidney transplant recipients (PRA ≥ 98%) in the OPTN/UNOS database into two distinct clusters that demonstrated high stability. This study identified that the two distinct subtypes had different clinical outcomes, specifically in relation to acute rejection and death-censored graft failure.

Mechanisms for sensitization are likely to be different between the two clusters as the cluster 1 recipients were more likely to be male and undergoing re-transplantation while the cluster 2 recipients were more likely to be female and undergoing a first-time transplantation. The first cluster, cluster 1, was more likely to be composed of male recipients who were undergoing re-transplantation, and their immune system may have recognized the new transplanted organ as foreign and mounted a response against it, potentially leading to rejection. In contrast, the second cluster, cluster 2, was more likely to be composed of female recipients undergoing their first transplant. These individuals may have been sensitized by prior pregnancies, which can lead to the development of antibodies against the transplanted organ as foreign and lead to rejection. Additionally, more recipients in cluster 1 had a PRA of 100%, suggesting that their immune system had a high level of pre-existing antibodies due to the sensitizing events associated with their prior transplants, which could have led to the development of a more robust immune response. In summary, the two clusters of organ transplant recipients may have had different mechanisms for sensitization based on their gender, previous transplant history, or pregnancies, which could affect their likelihood of developing rejection after transplant.

Furthermore, cluster 1 recipients were also more likely to have had exposure to prior immunosuppression, which is associated with an increased risk of infection and malignancy, in addition to the risk of donor-specific antibody reactivation and antibody-mediated graft rejection. These cumulatively may have translated to cluster 1′s lower patient and graft survival [1,24,25,26,27]. It has been acknowledged that the mode of sensitization can predict graft survival. Sensitization due to prior pregnancies or transfusions has been found to increase the risk of graft loss by 23%, while re-transplantation increases the risk by 58% [28].

Recipients in cluster 1 had a higher likelihood of receiving dialysis for over three years before transplantation compared to those in cluster 2. This prolonged waiting time and dialysis vintage were likely contributors to cluster 1’s inferior long-term outcomes [29,30,31]. Prior research has indicated that patients who underwent preemptive transplants exhibited superior graft and patient survival rates than those with dialysis vintage durations of <5 years, 5–9 years, and ≥10 years [29]. Cold ischemia time was also found to be associated with a higher risk of delayed graft function; however, this effect was modest and had less impact than the kidney donor profile index (KDPI). Therefore, prolonged cold ischemia time alone should not be considered as a primary reason to decline transplantation [30]. Unexpectedly, cluster 1 recipients had less diabetes (17% vs. 35%, respectively) and diabetic-related ESKD (3% vs. 29%, respectively) compared to cluster 2. Of note, ESKD was caused by unknown factors in around 70% of cluster 1 patients compared to only 16% in cluster 2. Although more patients in cluster 2 had diabetes, improvements in diabetes care and the optimization of immunosuppressant therapy may contribute to their superior graft outcomes.

There has been a long-standing debate in the transplant community about the appropriate level of priority that should be given to highly sensitized patients. The current algorithm allocates kidneys based on the level of PRA, which some have argued gives too much weight to sensitization status. In the present study, the majority of kidney transplants in both clusters 1 and 2 were from standard KDPI non-ECD donors, indicating that these patients had access to good quality allografts. However, the transplant outcomes observed in highly sensitized patients were not as good as might be expected. Recent discussions about the development of a new kidney allocation framework have focused on decreasing the emphasis on PRA and providing more equitable access to kidney transplantation for patients on the waiting list. These discussions have been fueled in part by the findings of studies like this one, which highlight the need for a more nuanced approach to kidney allocation. Specifically, the study suggests that while PRA remains an important factor in determining priority for transplantation, other factors such as dialysis vintage and waiting time should also be taken into account [32]. Additionally, the study underscores the need for further exploration of the access and priority concerns for highly sensitized patients. Certain very highly sensitized patients received preemptive transplants while others did not, raising questions about the effectiveness of the current allocation system in addressing the diverse sensitization statuses of patients. A comprehensive kidney allocation framework that considers multiple factors such as PRA, waiting time, dialysis vintage, and access will be critical to ensuring equitable and unbiased access to kidney transplantation for all patients.

In this study, there were several limitations that should be acknowledged. First, due to the national registry’s limitations, details regarding graft rejection, graft failure, and patient death are lacking. Moreover, there might be missing data and loss of follow-up in patients, which could have led to the underestimation of outcomes. However, to minimize bias, we utilized the multivariable imputation by the chained equation strategy for missing data. Additionally, the OPTN/UNOS registry tends to underreport rejection events, so the reported rejection events in this study may not fully represent the immunologic risk events in highly sensitized patients. Finally, our outcomes are not reflective of highly sensitized patients who remain on the waitlist.

To our knowledge, this is the first ML clustering approach that has been successfully applied to very highly sensitized kidney transplant recipients (PRA ≥ 98%). The ML clustering algorithms allowed for the identification of two distinct subgroups without human intervention. The patients in cluster 2 had better outcomes regarding death-censored graft survival and acute rejection, indicating that highly sensitized kidney transplant recipients, even with a PRA of ≥98%, are a heterogeneous population. Consequently, this categorization could enable targeted interventions to improve outcomes. Furthermore, the different cluster distributions among the 11 OPTN/UNOS regions may facilitate the identification of future management strategies that incorporate geographical location to enhance outcomes for highly sensitized kidney transplant recipients.

While this study’s unsupervised ML clustering approach provides extensive insight into the various phenotypes of very highly sensitized kidney transplant recipients in the United States and their corresponding post-transplant outcomes, it is important to note that ML clustering algorithms have limitations in terms of directly generating risk predictions for individual cases. As such, future research should investigate the efficacy of supervised ML prediction models (such as neural network and extreme gradient boosting) utilizing labeled outcomes for the purpose of predicting graft loss and mortality in very highly sensitized kidney transplant recipients. These models should be compared against traditional prediction models and standard risk-adjusted outcomes to ascertain their predictive performances.

## 5. Conclusions

Our study successfully utilized an unsupervised ML consensus clustering tool to classify highly sensitized kidney transplant recipients in the OPTN/UNOS database into two subtypes with differing clinical outcomes, specifically acute rejection and death-censored graft loss. The results of our study demonstrate that consensus clustering is a viable and effective method for identifying patient subgroups with distinct clinical characteristics in very highly sensitized kidney transplant patients. By tailoring treatment plans based on subgroup membership, this approach may enhance the clinical outcomes and inform clinical decision-making.

## Figures and Tables

**Figure 1 medicina-59-00977-f001:**
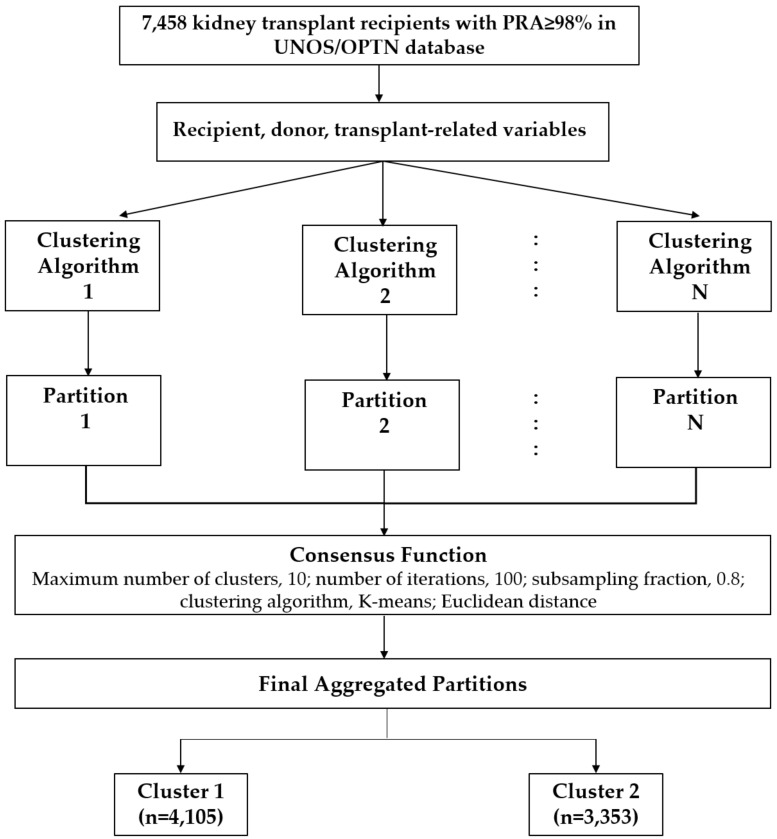
Workflow of consensus clustering. Abbreviations: OPTN: Organ Procurement and Transplantation Network, PRA: Panel reactive antibody, UNOS: United Network for Organ Sharing.

**Figure 2 medicina-59-00977-f002:**
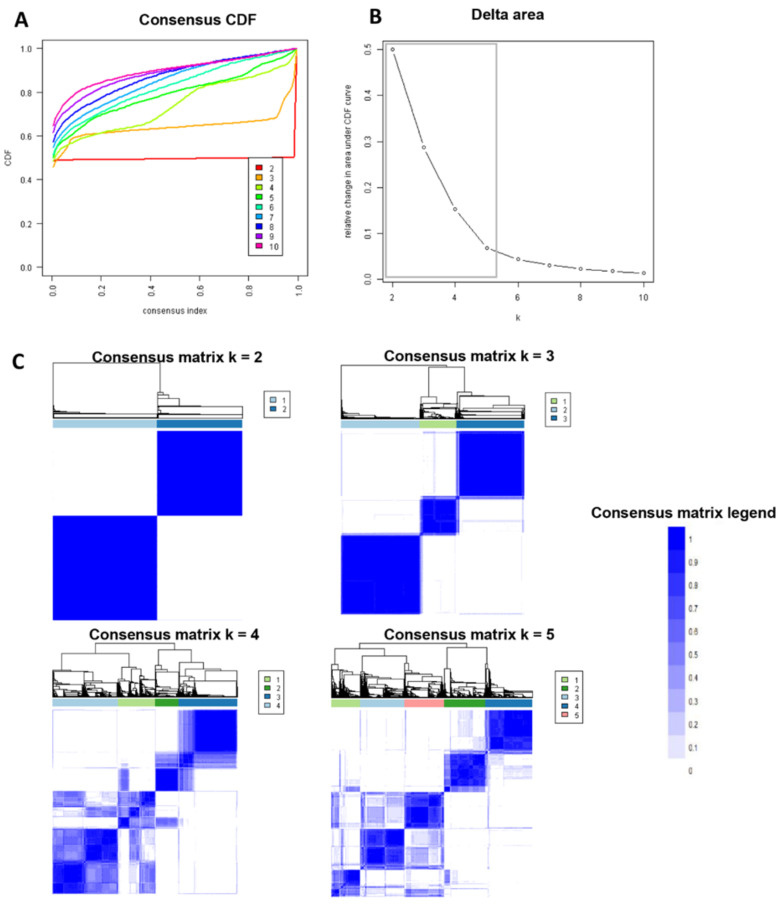
Displays three visual representations of the consensus clustering analysis for highly sensitized kidney transplant patients. Panel (**A**) shows a cumulative distribution function (CDF) plot, which displays the distribution of consensus values for each potential number of clusters (k) that were tested. Panel (**B**) shows a delta area plot, which represents the relative increase in the area under the CDF curve as the number of clusters increases from k = 2 to k = 10. Panel (**C**) shows a consensus matrix (CM) heat map, which depicts the consensus values for each pair of patients within each identified cluster, represented by shades of blue on a white-to-blue color scale.

**Figure 3 medicina-59-00977-f003:**
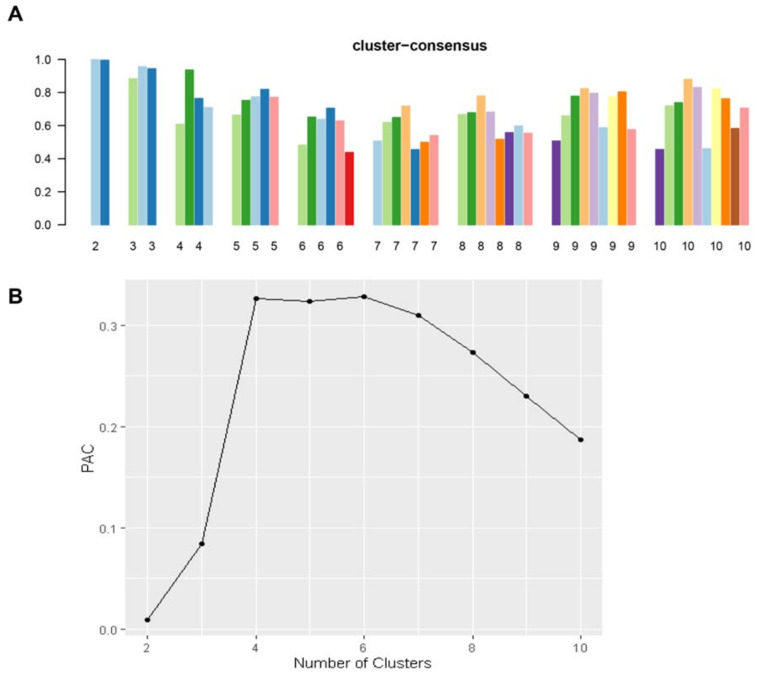
Displays two graphs. The first graph (**A**) is a bar plot that shows the mean consensus score for different numbers of clusters (k ranges from two to ten). Different colors indicate different cluster group. The second graph (**B**) depicts the PAC values that assess ambiguously clustered pairs.

**Figure 4 medicina-59-00977-f004:**
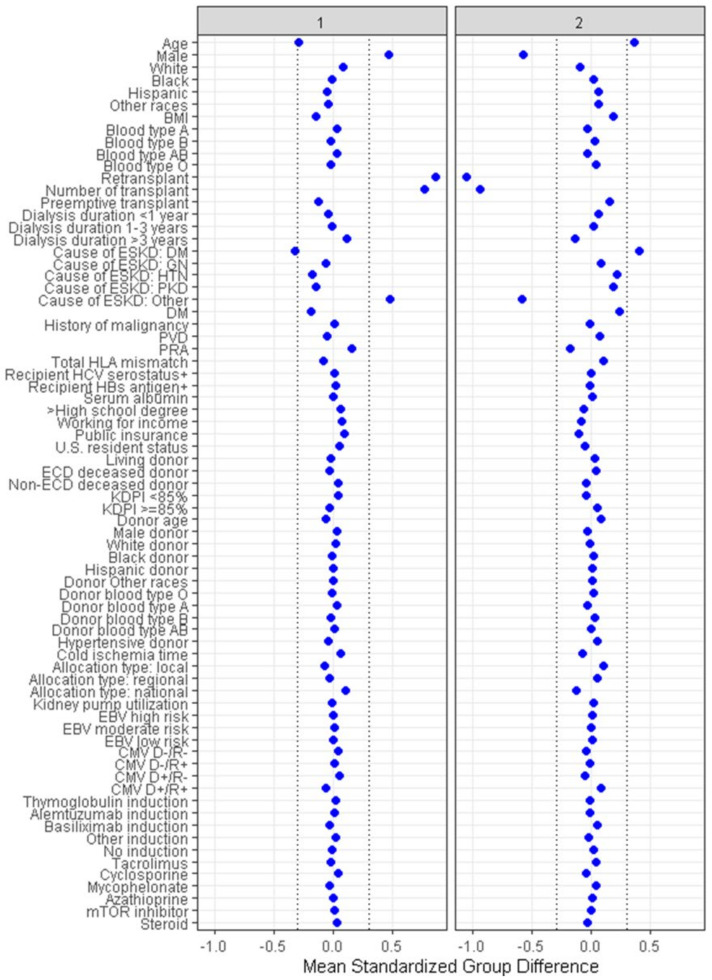
Displays a plot of the standardized mean differences, which identifies the clinical characteristics of each cluster.

**Figure 5 medicina-59-00977-f005:**
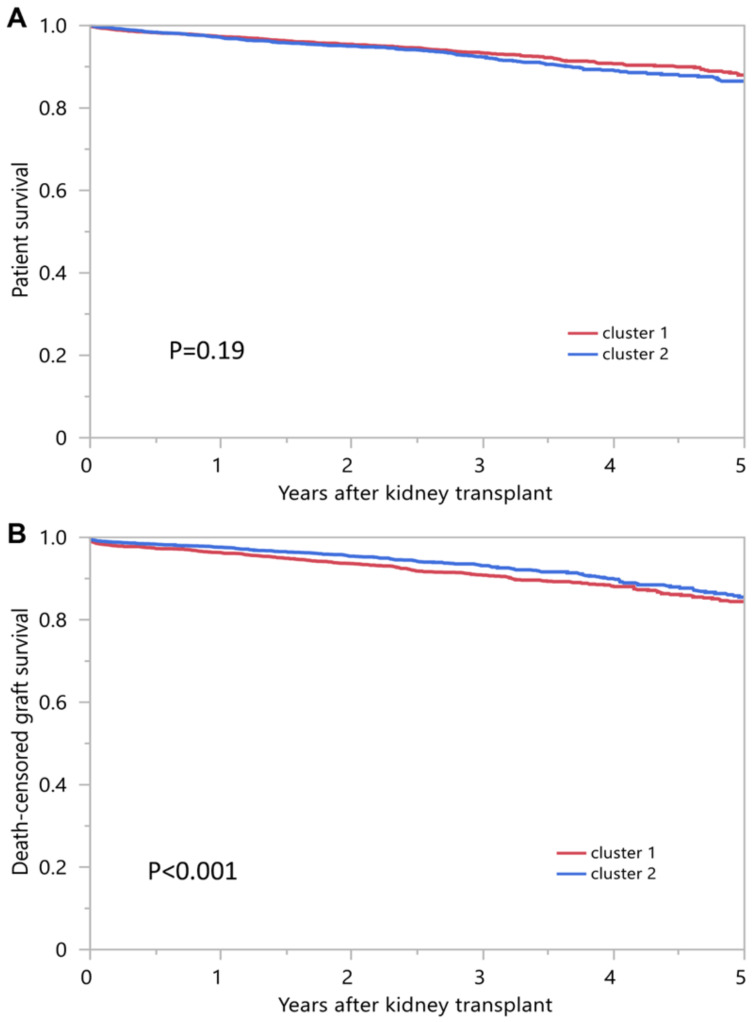
Displays the patient survival (**A**) and death-censored graft survival (**B**) outcomes for the two distinct clusters of very highly sensitized kidney transplant recipients identified by consensus clustering analysis.

**Table 2 medicina-59-00977-t002:** Post-transplant outcomes according to the clusters.

	Cluster 1	Cluster 2
1-year survival	97.3%	97.1%
HR for 1-year death	0.93 (0.69–1.24)	1 (ref)
5-year survival	88.0%	86.5%
HR for 5-year survival	0.88 (0.73–1.06)	1 (ref)
1-year death-censored graft survival	96.3%	97.6%
HR for 1-year death-censored graft failure	1.57 (1.18–2.09)	1 (ref)
5-year death-censored graft survival	84.4%	85.5%
HR for 5-year death-censored graft failure	1.28 (1.07–1.54)	1 (ref)
1-year acute rejection	9.2%	5.4%
OR for 1-year acute rejection	1.78 (1.48–2.14)	1 (ref)

Abbreviations: HR: hazard ratio; OR: odds ratio.

## Data Availability

Upon reasonable request, the authors are willing to share the data.

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
