# Peer review of "Differences between Very Highly Sensitized Kidney Transplant Recipients as Identified by Machine Learning Consensus Clustering"

_medicina, 2023, doi:10.3390/medicina59050977_

Round 1

Reviewer 1 Report

In this study authors categorized very highly sensitized kidney transplant recipients using an unsupervised machine learning approach. They found two distinct groups of patients according to a difference in outcomes. This method is new and it may be helpful for a better understanding of most important clinical factors related to worse outcomes in kidney transplantation.

I am not an expert of machine learning technique but as a clinician I do not see a significant clinical difference in outcomes, when a difference of death-censored graft survival was 1.3% in one year and 1.1% in 5 years. I do agree that a difference in 1-year acute rejection rate may be counted as clinically different though authors state that graft rejection episodes were underreported.

Would it be helpful to combine unsupervised learning method with supervised learning and consensus clustering in order to get more clear differentiation? Or to aim for a 5 year death-censored graft survival difference between the groups of at least 10%?

If improvement can not be made using this technique I would suggest to make some explanations in methods and corrections of conclusions.

Author Response

 Response to Reviewer#1

Comment #1

In this study authors categorized very highly sensitized kidney transplant recipients using an unsupervised machine learning approach. They found two distinct groups of patients according to a difference in outcomes. This method is new and it may be helpful for a better understanding of most important clinical factors related to worse outcomes in kidney transplantation.

I am not an expert of machine learning technique but as a clinician I do not see a significant clinical difference in outcomes, when a difference of death-censored graft survival was 1.3% in one year and 1.1% in 5 years. I do agree that a difference in 1-year acute rejection rate may be counted as clinically different though authors state that graft rejection episodes were underreported.

Would it be helpful to combine unsupervised learning method with supervised learning and consensus clustering in order to get more clear differentiation? Or to aim for a 5-year death-censored graft survival difference between the groups of at least 10%?

If improvement cannot be made using this technique I would suggest to make some explanations in methods and corrections of conclusions.

Response: Thank you for your review and critical evaluation. We agree with the reviewer’s important point and thus we additionally make explanations as the reviewer’s suggestion. While this study's unsupervised ML clustering approach provides extensive insight into the various phenotypes of very highly sensitized kidney transplant recipients in the United States and their corresponding posttransplant outcomes, it is important to note that ML clustering algorithms have limitations in terms of directly generating risk predictions for individual cases. As such, future research should investigate the efficacy of supervised ML prediction models (such as neural network and extreme gradient boosting) utilizing labeled outcomes for the purpose of predicting graft loss and mortality in very highly sensitized kidney transplant recipients following transplantation. These models should be compared against traditional prediction models and standard risk-adjusted outcomes to ascertain their effectiveness. This key point, suggested by the reviewer, has been incorporated into our discussion.

While this study's unsupervised ML clustering approach provides extensive insight into the various phenotypes of very highly sensitized kidney transplant recipients in the United States and their corresponding posttransplant outcomes, it is important to note that ML clustering algorithms have limitations in terms of directly generating risk predictions for individual cases. As such, future research should investigate the efficacy of supervised ML prediction models (such as neural network and extreme gradient boosting) utilizing labeled outcomes for the purpose of predicting graft loss and mortality in very highly sensitized kidney transplant recipients. These models should be compared against traditional prediction models and standard risk-adjusted outcomes to ascertain their performances.

Thank you for your time and consideration.  We greatly appreciated the reviewer's and editor's time and comments to improve our manuscript. The manuscript has been improved considerably by the suggested revisions.

Reviewer 2 Report

The authors used different unsupervised machine learning algorithms to classify the sensitized-24 kidney transplant recipients with pre-transplant panel reactive antibody (PRA) ≥98%. The results of the work demonstrated that consensus clustering is a viable and effective method for identifying patient subgroups with distinct clinical characteristics in very highly sensitized kidney transplant patients

(1) The authors are suggested to provide the details of data collection and prepossessing and draw necessary example figures to clearly show the key steps of the used method.

(2) Proofreading is suggested.  Some grammar errors and improper phrases are found.

Author Response

Response to Reviewer#2

The authors used different unsupervised machine learning algorithms to classify the sensitized-24 kidney transplant recipients with pre-transplant panel reactive antibody (PRA) ≥98%. The results of the work demonstrated that consensus clustering is a viable and effective method for identifying patient subgroups with distinct clinical characteristics in very highly sensitized kidney transplant patients

Response: Thank you for your review and critical evaluation

Comment #1

The authors are suggested to provide the details of data collection and prepossessing and draw necessary example figures to clearly show the key steps of the used method.

Response: The following statements have been added into the method.

“OPTN/UNOS database contains patient-level data of all transplant events in The United States. Recipient-, donor-, and transplant-related variables, listed in Table 1, were abstracted from the OPTN/UNOS database to be included in the ML cluster analysis. All variables had less than 10% missing data (Table S1), and missing data were imputed using the multivariable imputation by the chained equation (MICE) method.”

We also added figure 1 to display the key steps and workflow of consensus clustering.

Comment #2

Proofreading is suggested.  Some grammar errors and improper phrases are found.

Response: The manuscript has been proofread as suggested.

Thank you for your time and consideration.  We greatly appreciated the reviewer's and editor's time and comments to improve our manuscript. The manuscript has been improved considerably by the suggested revisions.
